# Platelet Contribution and Endothelial Activation and Stress Index-Potential Mortality Predictors in Traumatic Brain Injury

**DOI:** 10.3390/ijms25147763

**Published:** 2024-07-16

**Authors:** Alexandru Emil Băetu, Liliana Elena Mirea, Cristian Cobilinschi, Ioana Cristina Grințescu, Ioana Marina Grințescu

**Affiliations:** 1Department of Anesthesiology and Intensive Care II, Carol Davila University of Medicine and Pharmacy, 050474 Bucharest, Romania; alexandru.baetu@rez.umfcd.ro (A.E.B.); ioana.grintescu@umfcd.ro (I.M.G.); 2Department of Anesthesiology and Intensive Care, Grigore Alexandrescu Clinical Emergency Hospital for Children, 011743 Bucharest, Romania; 3Department of Anesthesiology and Intensive Care, Clinical Emergency Hospital Bucharest, 014461 Bucharest, Romania; 4Department of Anesthesiology and Intensive Care, Zetta Clinic, 020311 Bucharest, Romania; ioana_grintescu@yahoo.com

**Keywords:** traumatic brain injury, trauma-induced coagulopathy, TBI-induced coagulopathy, PLTEM, EASIX, ROTEM, clot elasticity

## Abstract

Coagulopathy and traumatic brain injury (TBI) are complexly intertwined. In isolated TBI, coagulopathy may contribute to hemorrhagic lesion development, progression, or recurrence, as it may lead to a particular pattern of coagulopathy called TBI-induced coagulopathy (TBI-IC). We performed a retrospective and descriptive evaluation of 63 patients admitted to the Emergency Clinical Hospital Bucharest with the diagnosis of moderate/severe brain injury. In addition to demographic data, all included patients had a complete paraclinical evaluation that included rotational thromboelastometric (ROTEM) blood-clot analysis. The platelet component (PLTEM) and the endotheliopathy activation and stress index score (EASIX) were calculated. These parameters were presented comparatively according to survival at 30 days and helped define the two study groups: survivors and non-survivors at 30 days. The contribution of platelets to clot strength is derived from maximum clot elasticity (MCE) and maximum clot firmness (MCF). MCE is defined as (MCF × 100)/(100 − MCF), and PLTEM is defined as EXTEM MCE—FIBTEM MCE. EASIX is a novel biomarker recently studied in TBI patients, calculated according to the following formula: lactate dehydrogenase (U/L) × creatinine (mg/dL)/platelets (10^9^ cells/L). Regarding the demographic data, there were no significant differences between the survivors and non-survivors. All ROTEM parameters related to clot amplitude (A5, A10, A20, MCF in EXTEM and FIBTEM channels) were higher in the group of patients who survived. Also, PLTEM was decreased in the group of deceased patients (89.71 ± 22.86 vs. 132.3 ± 16.56 *p* < 0.0001). The cut-off point determined with the ROC curve is 114.10, with a sensitivity of 94.74% and a specificity of 93.18%, for the detection of the negative prognosis (death at 30 days). The EASIX score was significantly higher in the patients who survived the traumatic event, with a median difference value of 1.15 (*p* < 0.0001). The ROC analysis of this biomarker highlights a cut-off point of 2.12, with a sensitivity of 88.64% and a specificity of 94.74% (AUC = 0.95, *p* < 0.0001), for the prediction of mortality. The comparative analysis of the two studied markers was performed using the Cox proportional hazard ratio and highlighted the greater influence that PLTEM has on survival time (b value = −0.05, *p* < 0.0001) compared to EASIX (b value = 0.49, *p* = 0.0026). The present retrospective study indicates the potential of the TBI-IC reflecting parameters PLTEM and EASIX as markers of mortality prognosis. Larger prospective studies are needed to confirm their combined prognostic value and use in decision-making and reduction in the burden of disease by adequate allocation of resources in a personalized and timely manner.

## 1. Introduction

According to the latest World Health Organization Global Burden of Disease Study, globally, in 2019, there were 27.16 million new cases of traumatic brain injury (TBI), 48.99 million prevalent cases, and 7.08 million years lost to disability (YLDs); more than 60 thousand TBI-related deaths were recorded in the USA alone in the same year [1]. According to a systematic analysis of the global, regional, and national burden of traumatic brain injury and spinal cord injury, although declining since 1990, the high burden of TBI remains a concern in all countries; therefore, prediction in TBI is of pivotal importance for healthcare personnel and policymakers [2]. Mortality prediction for TBI was studied extensively using both traditional and machine-learning algorithms, and the five strongest traditional markers for mortality prediction were the Glasgow coma scale (GCS), age, pupillary light reflex, injury severity score (ISS) for brain region, and the presence of acute subdural hematoma [3]. The mortality of TBI patients with coagulopathy varies from 17 to 86% [4,5]. A deeper, more recent understanding of the highly elaborate processes that develop after TBI highlights the role of coagulopathy as a prognostic marker [4,6,7,8].

Coagulopathy and TBI are complexly intertwined. In isolated TBI, coagulopathy may contribute to hemorrhagic lesion development, progression, or recurrence, as it may lead to a particular pattern of coagulopathy called TBI-IC [4,9,10,11]. TBI-IC is a systemic response to a localized injury. In the early stages, it is characterized by hypocoagulability due to hyperfibrinolysis and platelet dysfunction, which translates clinically into a disseminated intracranial hemorrhage, delayed intracranial or intracerebral hematoma formation, systemic bleeding, and, later on, by the increased risk of local and systemic thrombosis [9,12]. In complex multiple-trauma patients, secondary cerebral lesions may be precipitated by trauma-induced coagulopathy (TIC), a distinct hemorrhagic derangement from that of TBI-IC. The two coagulation profiles, TIC and TBI-IC, may coexist, thus making the diagnosis and treatment of coagulopathy particularly challenging in TBI patients.

The mechanisms of TBI-IC include direct effects of injury, such as microvascular failure and blood–brain barrier (BBB) disruption that lead to subsequent platelet–endothelium interactions and platelet exhaustion, exposure of brain tissue factor (TF) to clotting factors—in particular, to factor VIIa, and release of endogenous tissue-type and urokinase-type plasminogen activators (tPA and uPA) [4]. Patients with TBI-IC develop, early in their course, a hyperfibrinolytic state, raising the question of whether TBI-induced hyperfibrinolysis is triggered, accelerated, and enhanced by fibrin formed on the surface of microvesicles released after BBB breach and initiated independently of fibrin formation. Accordingly, elevated plasma tPA and fibrin degradation products are associated with progressive hemorrhagic injury in patients with TBI [13].

Whenever activation of the sympathetic nervous system occurs as a consequence of injury, it triggers vasoconstriction, activation of endothelial cells, capillary leakage, formation of a procoagulant microenvironment, local edema, microthrombosis, and activation of immune cells, the sum of which constitute shock-induced endotheliopathy (SHINE) and lead to organ failure and increased mortality [14]. Resuscitation-associated endotheliopathy (RAsE) and subsequent coagulopathy, mostly through aggressive volume-expansion resuscitation, may further aggravate SHINE [15]. EASIX is a novel biomarker that was initially designed to evaluate the severity of endotheliopathy after stem-cell transplantation and has recently been studied in TBI patients with promising results [16].

Qualitative and quantitative platelet deficits appear to be significant contributors to TBI-IC and are associated with an increased risk of developing complications. Brain-derived platelet-activating factor (PAF) also plays a role in BBB breakdown and promotes the release of additional PAF, TF, and other procoagulant molecules, such as brain-derived cellular microvesicles (BDMVs), which lead to platelet hyperactivity followed by platelet depletion and further hemorrhagic complications. Adenosine diphosphate (ADP) and arachidonic acid (AA) platelet receptor inhibition is a distinct yet common feature of TBI-IC; occurs in the absence of shock, hypoperfusion, or TIC; and does not appear to be caused by granule depletion of activated platelets. This underlines that TBI alone is sufficient to induce significant platelet dysfunction [9,17].

Together with the conventional coagulation assay (CCA), platelet function tests and viscoelastic hemostatic assays (VHA) have been used to better characterize the dynamic processes in TBI-IC to guide the administration of erythrocytes, fresh frozen plasma, platelet concentrate, coagulation factor concentrates, antifibrinolytics, platelet function enhancers, reversal agents, and antithrombotics in the later stages and also to predict clinical evolution, since coagulopathy is a strong predictor of mortality in TBI patients [6,12,18,19,20,21]. Modified VHA with platelet function assays or whole-blood platelet multiple electrode aggregometry (MEA) tests have further helped understand the underlying TBI-IC pathophysiology and guide clinical decisions [22]. Analysis of the TBI subgroup of patients from two recent trials conducted by Gonzalez et al. and Baksaas-Aasen et al. demonstrated a significant 28-day mortality benefit with VHA-guided management versus CCA [21,23,24,25,26]. Another study showed that ROTEM-guided coagulation management in patients with isolated TBI who underwent craniotomies led to improved clot quality and a decreased incidence of progressive hemorrhagic injury and neurosurgical reintervention [27].

In theory, the platelet contribution to clot formation could be derived with the aid of PLTEM—the ROTEM platelet component obtained by subtracting FIBTEM MCF (FIBTEM channel uses cytochalasin D, an agent that blocks platelet activity, thus reflecting only fibrin network/fibrinogen contribution) from EXTEM MCF (EXTEM channel assesses both fibrin network/fibrinogen and platelet contribution to clot formation). However, from a physical point of view, the clot that undergoes rotational thromboelastometry analysis is subject to oscillometric stress that results in a molecular transposition known as “creep”, which influences viscosity. The extrapolation of clot properties from viscoelastic materials initially suggested that a blood clot behaves as a Maxwell body, a physical model that takes into account a material’s viscoelastic properties in relation to stress, strain, and changes in these parameters over time. However, recent animal studies indicate that the Zener model or standard linear solid model may be more adequate since they also include a description of molecular creep [28]. When stress is removed, the clot’s elasticity returns to its original form. Consequently, the specific assessment of the contribution of platelets to clot strength can be derived from the elasticity results—MCE. It is essential that the platelet component is determined using elasticity as opposed to clot amplitude because of the nonlinear relationship between the two, in accordance with clotting physiology and the laws of physics. MCE is defined as (MCF × 100)/(100 − MCF); hence, ROTEM platelet component PLTEM is derived as EXTEM MCE—FIBTEM MCE [28].

The addition of PLTEM to classical ROTEM-guided transfusion protocols was used in assessing platelet contribution to coagulopathy in trauma, adult, pediatric, and neonatal cardiac surgery; post-cardiac bypass; and postpartum hemorrhage [28,29,30,31,32,33]. To our knowledge, PLTEM has not been studied in TBI and TBI-IC [22,34]. We hypothesized that the ROTEM-derived marker PLTEM together with EASIX could be used as dynamic tools of TBI-induced coagulopathy and mortality prognosis instruments in TBI patients [22,35,36].

## 2. Results

The present study includes 63 patients of whom 39 are men and 24 are women. The mean age of the entire study group is 42.32 years, with a standard deviation of 16.40 years. The cause of cerebral trauma was represented mainly by traffic accidents (n = 34, 53.97%), followed by assaults (n = 19, 30.16%) and falls from a height in ten cases (15.87%). A comparative analysis of the study group data divided by first 30-day survival revealed similar ages and body mass index. From the TBI severity perspective, the surviving group is represented by 30 severe TBI (77.27%) and 10 moderate TBI patients (22.73%), and the group of deceased patients comprises 16 severe TBI (84.21%) and 3 moderate TBI (15.79%) cases. A comparative analysis of distribution frequencies with the Fisher’s exact test did not reveal a statistically significant difference (*p* = 0.73). The same test was used in analyzing the lesional pattern frequency of distribution, which was consistent with no statistically significant difference between the two groups, with subarachnoid hemorrhage (*p* = 0.78) being the culprit in the majority of cases, followed by subdural hematoma (*p* = 0.13) and epidural hematoma (*p* = 0.71). The analysis of the acid–base balance collected at admission emphasizes a statistically significant difference between the two groups regarding base excess-survivors and non-survivors (−7.03 ± 6.99 vs. −11.90 ± 10.85, *p* = 0.03). However, there are no differences in serum pH, lactate, and bicarbonate. Even though no differences were detected in hemoglobin and aPTT, the patients who died (median PLT = 108,000/μL) were more thrombocytopenic than those who survived (median PLT = 189,000/µL, *p* < 0.0001). Also, INR was more prolonged in the deceased patients (median INR = 2.4) compared to the survivors (median INR = 1.76, *p* = 0.02) (Table 1).

In Table 1, it can be observed that, in the viscoelastic analysis of whole blood, there are major differences in terms of clot amplitude at 5, 10, and 20 min (A5, A10, A20) but also for MCF. Given that thrombelastometric analysis is dynamic, we performed the two-way ANOVA test to observe how the clot amplitude (at all times) changes quantitatively with the passage of time for the two groups studied (Figure 1 and Figure 2).

The variance detected for the EXTEM channel of the clot amplitude in the two groups is 26.59% (*p* < 0.0001), while the variance for the FIBTEM detected between the groups of deceased and surviving patients of the clot amplitude at 5, 10, and 20 min but also the MCF is 39.66% (*p* = 0.003) (Figure 1 and Figure 2).

The analysis of the derivatives of the clot elasticity on the two channels (EXTEM and FIBTEM) also pointed out a big difference between the survivors and the deceased (Figure 3 and Figure 4). On the EXTEM channel that takes into account both the fibrin network and platelets, the survivors had an average difference in clot elasticity of 47.23 (*p* < 0.0001) (150.3 ± 16.76 vs. 102.9 ± 23.55).

On the FIBTEM channel containing cytochalasin D (platelet inhibitor) as a reagent, a marked difference is observed between the group of survivors (18.01 ± 3.14) and that of the deceased patients (13.15 ± 3.84) with a high level of significance (*p* < 0.0001).

The analysis of the platelet contribution to the blood clot derived from the difference in clot elasticity on the two thromboelastometric channels studied (EXTEM and FIBTEM) highlights a marked reduction in the group of patients who died compared to those who survived the medium–severe brain traumatic event (89.71 ± 22.86 vs. 132.3 ± 16.56, *p* < 0.0001) (Figure 5). The cut-off point determined with the ROC curve is 114.10, with a sensitivity of 94.74% and a specificity of 93.18%, for the detection of the negative prognosis (death) of the patients in the first 30 days after the traumatic event (AUC = 0.93, *p* < 0.0001) (Figure 6).

When comparing the EASIX score, a major difference was observed between the group of survivors (median value = 1.15, 25% percentile = 0.80, and 75% percentile = 1.56) and the group of deceased patients (median value = 3.91, 25% percentile = 2.69, and 75% percentile = 1.56). The difference between the medians has a high level of statistical significance (*p* < 0.0001). The ROC analysis of this biomarker highlights a cut-off point of 2.12, with a sensitivity of 88.64% and a specificity of 94.74% (AUC = 0.95, *p* < 0.0001), for the prediction of mortality (Figure 7 and Figure 8).

Given the very high values of the areas under the curve (AUC) and the high predictive powers of death at 30 days for PLTEM and EASIX, we performed a Hanley and McNeil analysis to compare them. The difference between the areas under the curve is 0.015 (PLTEM AUC = 0.93 and EASIX AUC = 0.95, z = 0.31, *p* = 0.75) (Figure 9).

The Spearman analysis revealed an inverted correlation between PLTEM and EASIX objectified by a value of the correlation coefficient r of −0.57 (95% CI −0.72 to −0.37, *p* < 0.0001). Practically, the greater the platelet contribution, the lower the endothelial stress (Figure 10).

We performed a Cox proportional hazard ratio regression analysis using 30-day survival as the endpoint in order to simultaneously evaluate the effect of the possible predictive variables (PLTEM and EASIX) and highlight how our derived markers influence the hazard of death at 30 days.

The greatest influence on the hazard of death after moderate/severe TBI in our study is the platelet-derived component PLTEM (b = −0.05, Wald value = 18.80, *p* < 0.0001), followed by EASIX (b = 0.49, Wald value = 9.08, *p* = 0.0026). The regression model had a null-model fit (Log-Likelihood) of 151,723, a full-model fit (Log-Likelihood) of 97.72, and Chi-squared = 53.99 (*p* < 0.0001).

## 3. Discussion

Progress has been made in reducing the global burden of TBI in recent years. However, it remains one of the leading causes of YLDs globally. We are in a constant search for markers that could be addressed to reduce mortality and morbidity in a timely manner, and among these, coagulopathy seems to play a central role. Specific markers of coagulopathy are derived from the actual understanding of complex physiological processes and are being investigated for diagnostic and prognostic value [1,2,3].

Prognostication is essential in giving families and healthcare personnel reasonable expectations that aid in decision-making in severe, debilitating diseases, such as TBI. Simultaneously, prognostication helps reduce the global burden of disease by adequate allocation of resources in a personalized manner. We demonstrated that specific thromboelastometry-derived parameters that reflect platelet contribution to clot formation, PLTEM, and the markers of endotheliopathy, EASIX, mirror essential aspects of TBI-IC physiopathology and could be used to create a TBI-IC prognosis tool. Further prospective studies conducted in larger centers are needed to investigate a combined score for mortality prediction in TBI using PLTEM and EASIX.

Increased patient age and the proportion of concomitant use of platelet inhibitors, anticoagulants, and other drugs; plant-based supplements; or devices that influence normal blood clotting before any lesion occurs, as well as resuscitation-induced coagulopathy also contribute to altered hemostatic profiles, and they need to be carefully assessed and addressed in the management of TBI patients. Coagulopathy is multifactorial, and the distinct pattern of TBI-IC is influenced by TIC, RAsE, subsequent resuscitation-associated coagulopathy, and pre-existent alterations of normal hemostasis that play a part in the intricate mechanisms of coagulation [12,14,15,37].

Isolated TBI-IC lacks classical causal factors for hemorrhagic shock and TIC, which suggests that TBI-IC differentiates itself from other patterns of coagulopathy due to brain-specific features. Some of the particularities are interactions between the disrupted BBB and plasma proteins that result in endotheliopathy; exposure of large quantities of brain TF to clotting factors, which promotes a maladaptive activation of the extrinsic coagulation pathway with enhanced activation; insufficient inhibition and exacerbation of thrombin generation; subsequent exaggerated platelet activation leading to platelet consumption and exhaustion; hyperfibrinolysis due to the release of endogenous plasminogen activators and protein C pathway activation; complement system activation; and inflammation resulting in uncontrolled disseminated coagulation. Mixed bleeding and thrombotic phenotypes lead to an increased risk of secondary brain lesions and influence patients’ outcomes [5,9,38].

Endotheliopathy acts as a post-traumatic consequence of BBB disruption, contributing factor to TBI-IC, and systemic response to a localized injury. Specific pathways of TBI-induced endotheliopathy were described in animal models, including a model of “molecular memory” and alterations in gene expression likely responsible for distinct effects on the systemic microcirculation and cerebral vessels’ endothelium [39].

EASIX, as a marker of endothelial injury, was verified in different cohorts of patients with hematologic ailments after its validation in steroid-refractory graft-versus-host disease after allogeneic stem-cell transplantation as well as in COVID-19, sepsis, small-cell lung cancer, or advanced liver disease patients [16,40,41,42]. It has been used as an independent prognosis tool or in complex prognosis models but has not been studied sufficiently in TBI patients. A recent analysis of 358 TBI patients in a tertiary center showed EASIX was positively related to the mortality of TBI patients, and when investigating its prognostic value, EASIX had an AUC comparable to SOFA but lower than that of GCS. The AUC of GCS plus EASIX was improved compared with single GCS or EASIX. Furthermore, the AUC of a developed predictive model incorporating EASIX and the strongest traditional predictors for mortality—GCS, glucose, subdural hematoma, injury mechanism, and coagulopathy—was 0.874, with a sensitivity of 0.913 and specificity of 0.686 [16].

Thrombocytopenia is a strong negative prognostic factor in TBI, yet a normal platelet count could still be associated with a bleeding tendency due to multifactorial thrombocitopathy. Impaired platelet reactivity after trauma could be due to early, strong systemic activation of platelets, rendering platelets quiescent, decreased platelet adhesion and aggregation due to the shedding of adhesion receptor GPIbα and collagen receptor GPVI, altered von Willebrand factor–ADAMTS13 (a disintegrin and metalloprotease with thrombospondin type 1 motif 13) interactions, reduced aggregation due to hypofibrinogenemia, and impaired function under DAMPs (damage-associated molecular patterns) action. Particular TBI-IC qualitative alteration of platelet function occurs independent of shock, hypoperfusion, TIC, or granule depletion in activated platelets by cyclooxygenase pathway receptor inhibition [34,43].

We found PLTEM to be a marker of qualitative platelet contribution to clot formation or an image of thrombocytopathy, positively related with mortality in TBI patients. Accurate measurement of PLTEM using clot elasticity parameters is mandatory since it directly reflects an understanding of the oscillometric stress that influences the viscoelastic properties of the clot that undergoes rotational thromboelastometry analysis and subsequent derived data.

In our study, Cox proportional hazard ratio regression showed PLTEM to have the greatest influence on survival time after moderate/severe TBI, followed by EASIX, even though ROC curves were pretty similar. ROTEM-derived platelet contribution to clot formation PLTEM showed a great predictive value for mortality prediction (death in the first 30 days after injury), with a cut-off point of 114.11, 94.74% sensitivity, and 93.18% specificity. The predictive performance of EASIX was quite similar with PLTEM, with a cut-off point of 2.12, a sensitivity of 88.64%, and a specificity of 94.74%. EASIX integrates indirect markers of endothelial dysfunction, including platelet count, since low platelet levels may be attributable to hyperactivation and hyperaggregation.

EASIX embodies the endothelial component of coagulopathy itself but also the subsequent endotheliopathy-associated thrombocytopenia. From the best of our knowledge, this is the first study to confirm this pathophysiological link through a direct correlation analysis (r = −0.57, *p* < 0.0001) for moderate/severe TBI patients. It still remains to be established why PLTEM seems to better predict the hazard of death if the ROC curve for mortality prediction is similar to that of EASIX. A possible answer, which we cannot demonstrate in this unicentric study with a small number of patients, could be the possible involvement of PLTEM in the prediction of early deaths of moderate/severe TBI patients.

In trauma patients, a statistically significant but weak correlation has been reported between PLTEM and platelet count. Variability in the correlation between the two may be attributable to PLTEM being a measure of platelet function, which is thought to be distinct from platelet count, in particular, in the setting of coagulopathy [28,34]. A Swedish study assessing platelet function in TBI using MEA analysis showed that more than 60% of the patients exhibited pathologically low AA receptor values upon admission, but MEA values failed to independently correlate to outcome or lesion progression in multivariable analyses. Multiplate^®^, the ROTEM-adopted MEA platform, is not a VHA but an adjunct of ROTEM; thus, PLTEM and MEA results are not interchangeable [44].

There are several limitations to this study. We included a relatively small number of patients coming from the Emergency Clinical Hospital Bucharest, a single tertiary center that also serves as a referral center in Romania. Even if we excluded patients received by transfer from other hospitals more than six hours after the traumatic event, we could not avoid selection bias.

Complex prediction models are using multiple endpoints in neuro prognostication guidelines in critically ill patients with moderate/severe TBI; however, none include PLTEM and EASIX [45]. Survival was the only endpoint that we used in our study, but the promising results we have raised two main questions: whether these parameters could be integrated into multiple variable prognostication tools and whether they show the same relevance in more complex neurological outcome prediction for TBI (evolution to brain death, cerebral performance and cognitive status, functional outcome, quality of life, etc.), not only 30-day survival.

PLTEM and EASIX comprise a coagulation part as far as we know to date. Before we can guide a treatment or make any kind of recommendation, these derived markers will need extensive validation. Our conclusions should be verified in a larger number of TBI patients, ideally coming from more such centers. PLTEM was used to assess platelet contribution to coagulopathy in trauma and other clinical scenarios, but to our knowledge, it has not been studied in TBI and TBI-IC. Future studies could be designed to testify whether PLTEM could be used to reflect thrombocytopathy in TBI patients. Components of EASIX could be influenced by multiple unknown factors given the particular nature of TBI-induced endotheliopathy; therefore, classical markers of endothelial damage should be measured in order to certify the relationship between EASIX and the degree of endothelial injury in TBI patients.

## 4. Materials and Methods

We conducted a retrospective, descriptive study in which we included 63 patients admitted to the Emergency Clinical Hospital Bucharest in the last five years (starting from December 2019) with a diagnosis of moderate (GCS 9-12p) and severe (GCS 3-8p) TBI in which ROTEM analysis was performed. The abbreviated injury scale (AIS) score was a maximum of 5 (critical lesion) for the head and neck worst injury and a maximum of 2 (moderate lesion) in any other two body systems. Patients under chronic anticoagulant or antiplatelet treatment, under 18 years of age, who were pregnant, in which ROTEM analysis was not performed, suffering from multiple traumas with acute life-threatening lesions, and with chronic liver or kidney disease were subsequently excluded from the study. The Emergency Clinical Hospital Bucharest is a tertiary hospital, which has a department of neurocritical care. Therefore, it also receives patients with moderate and severe traumatic brain injury from other hospitals in Romania. Given the purpose of this study, we excluded patients who were admitted to our center by transfer from other hospital units more than six hours after the traumatic event.

Patients with a GCS of less than 8p were intubated either in the prehospital setting or immediately after admission for protection of the airway. All the patients were evaluated according to local protocols: complete physical examination, complete blood count, blood chemistry tests, arterial blood gas, conventional coagulation assay (prothrombin time (PT) values expressed as an international normalized ratio (INR)), activated partial thromboplastin time (aPTT)), and cerebral computed tomography. TBI patients were treated according to the latest European guidelines, with neuroprotection for prevention of secondary insults at the core of all interventions. Hemodynamic management was guided by intracerebral pressure (ICP) monitoring, and isotonic saline solution and vasoactive and cardioactive drugs were titrated to ensure normovolemia and optimal cerebral perfusion pressure. We targeted a blood glucose of 140–180 mg/dL, normoxemia, normocapnia, normothermia, and analgosedation. A rise in ICP of more than 22 mmHg compelled deepening of analgosedation, neuromuscular blockage, use of hyperosmolar solutions, a prudent increase in mean arterial pressure when autoregulation was preserved, cerebrospinal fluid drainage, or surgical exclusion of space-occupying lesions [46].

The following data were analyzed in this study: demographics—gender, age, weight, body mass index; paraclinical pH; base excess (BE); serum lactate (RAPIDPoint® 500 Blood Gas Systems - Siemens Healthineers, Erlangen, Germany); hemoglobin level (Hb); platelet count (Plt) (Celltac F Automated Hematology Analyzer - Nihon Kohden, Tokyo, Japan); INR; aPTT (ACL TOP 500- Werfen, Bedford, MA, USA), lactate dehydrogenase and serum creatinine (The ARCHITECT c4000 - Abbott, Lake Forest, IL, USA); thromboelastometry parameters—EXTEM channel (A5, A10, A20, MCF), FIBTEM channel (A5, A10, A20, MCF); derived elasticity parameters—EXTEM MCE, FIBTEM MCE; and information about survival at 30 days. We calculated the EASIX score according to the following formula: lactate dehydrogenase (U/L) × creatinine (mg/dL)/platelets (10^9^ cells/L) [16].

All patients had complete blood work analysis (including complete blood count, classical coagulation assay and ROTEM analysis) performed in the emergency department. Initially, if needed, patients received liberal blood transfusions, but after obtaining the results of the first thromboelastometric assay, they were switched to a goal-directed algorithm. In individual cases (severe coagulopathy, more than two hours delay between presentation and intensive care unit admission, or at the clinician’s discretion), the conventional coagulation assay and ROTEM were repeated at intensive care unit admission. We only used first, singular admission data for all the patients included in this study.

The VHA device used in this study was the ROTEM Sigma rotational thromboelastometer. Periodic quality control for monitoring the accuracy and precision of tests carried out on the ROTEM Sigma was performed as per the manufacturer’s instructions.

ROTEM Sigma (ROTEM^®^; TEM International, Munich, Germany) is a dynamic, real-time point of care VHA, which uses whole blood and allows for the measurement of global clot formation (initiation, amplification, propagation), clot strength, and clot dissolution kinetics by measuring and displaying the amount of a continuously applied rotational force transmitted to an electromechanical transduction system by the developing clot. More precisely, it consists of a cup containing whole-blood samples in which a pin suspended on a ball-bearing mechanism oscillates through the application of a constant force for a determined period of time. As the clot builds up and its strength increases, the rotation of the pin is hindered, and the oscillation angle decreases. Changes in clot strength are directly transmitted to a torsion wire, detected by a transducer, and graphically depicted. Thereby, a complete image of clot formation and clot lysis is made: clotting time (CT); dynamic clot amplitude at 5, 10, and 20 min (A5, A10, A20); maximum clot firmness (MCF); as well as clot lysis-lysis index at 30 min after CT (LI30). Clot amplitude is clinically translated as “clot strength” and is physiologically the interaction between platelets and the fibrin network [22,47].

ROTEM Sigma uses four working channels, each with a cup containing specific clotting activators: EXTEM—reagent contains recombinant tissue factor, calcium ions, and polybrene and provides information similar to that of PT; INTEM—reagent contains phospholipids, calcium ions, and ellagic acid and provides information similar to that of the aPTT; FIBTEM—reagent contains cytochalasin D, recombinant tissue factor, polybrene, and calcium ions and, when compared to EXTEM analysis, allows for the qualitative analysis of the fibrinogen contribution to clot strength independent of platelets; APTEM—reagent contains aprotinin or tranexamic acid, recombinant tissue factor, polybrene, and calcium ions and, when compared to EXTEM analysis, assesses fibrinolysis [47].

The statistical software used was GraphPad 10.2. Data normality was tested using the D’Agostino–Pearson test. Comparative analyses of paraclinical data for the study groups divided by the clinical outcome were performed with the *t*-test for independent variables and considered whether or not the Gaussian distribution was observed. The possible predictions of the analyzed ROTEM values for the subsequent evolution of the patient were tested using ROC curves. Cut-offs were generated according to the Youden index. Hanley and McNeil analysis was used to compare PLTEM and EASIX ROC curves with MedCalc 14.1. A comparative analysis of the impact of PLTEM and EASIX on survival was performed using the Cox proportional hazard ratio. The two-way ANOVA test was used to determine how the clot amplitude changes quantitatively in time for the two groups studied. In order to check the possible correlation between PLTEM and EASIX, we used the Spearman correlation coefficient.

This study was conducted in accordance with the Declaration of Helsinki and approved by the Clinical Emergency Hospital of Bucharest Ethics Committee (protocol code 11095/03.12.2019) for the data collection, analysis, and publishing of the results.

## 5. Conclusions

In conclusion, the present retrospective study indicates the potential of the parameters PLTEM and EASIX as markers of mortality prognosis, reflecting TBI-IC in patients with moderate and severe TBI. Larger prospective studies are needed to confirm their combined prognostic value. Currently, there are no high-quality recommendations regarding evaluation methods for TBI-IC, so the need to validate accessible bedside parameters is evident.

## Figures and Tables

**Figure 1 ijms-25-07763-f001:**
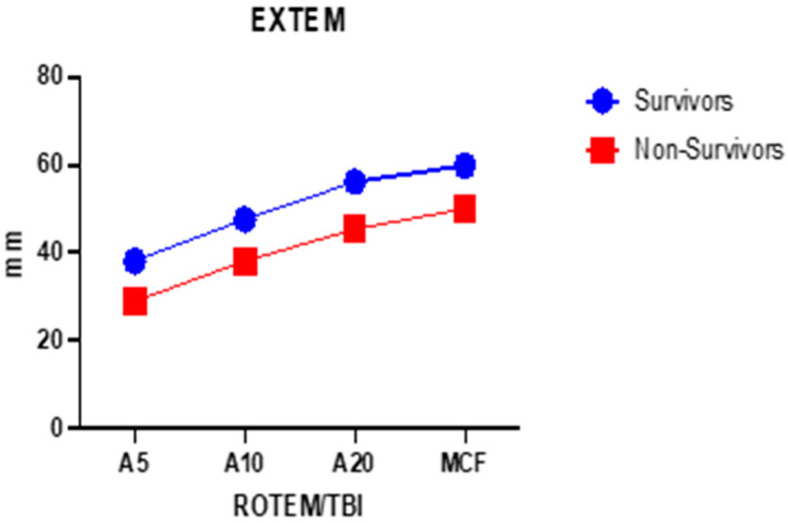
Mean values comparison of A5, A10, A20—clot amplitude in millimeters at 5, 10, and 20 min; and MCF—maximum clot firmness between survivors (n = 44) and non-survivors (n = 19), EXTEM channel.

**Figure 2 ijms-25-07763-f002:**
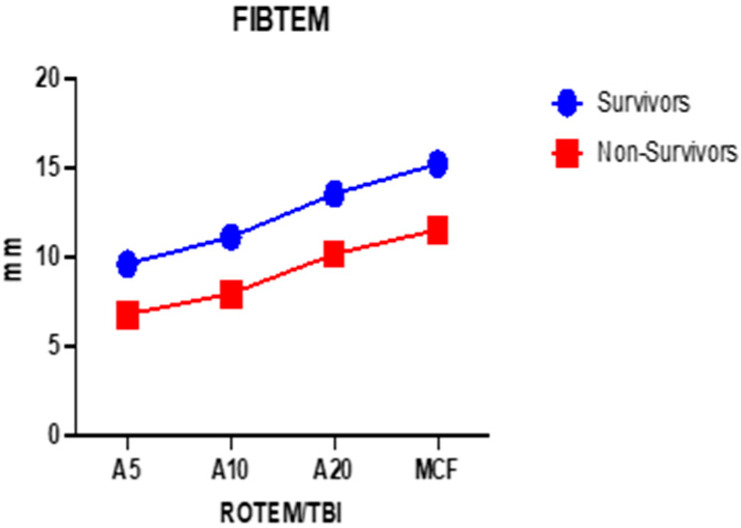
Mean values comparison of A5, A10, A20—clot amplitude in millimeters at 5, 10, and 20 min; and MCF—maximum clot firmness between survivors (n = 44) and non-survivors (n = 19), FIBTEM channel.

**Figure 3 ijms-25-07763-f003:**
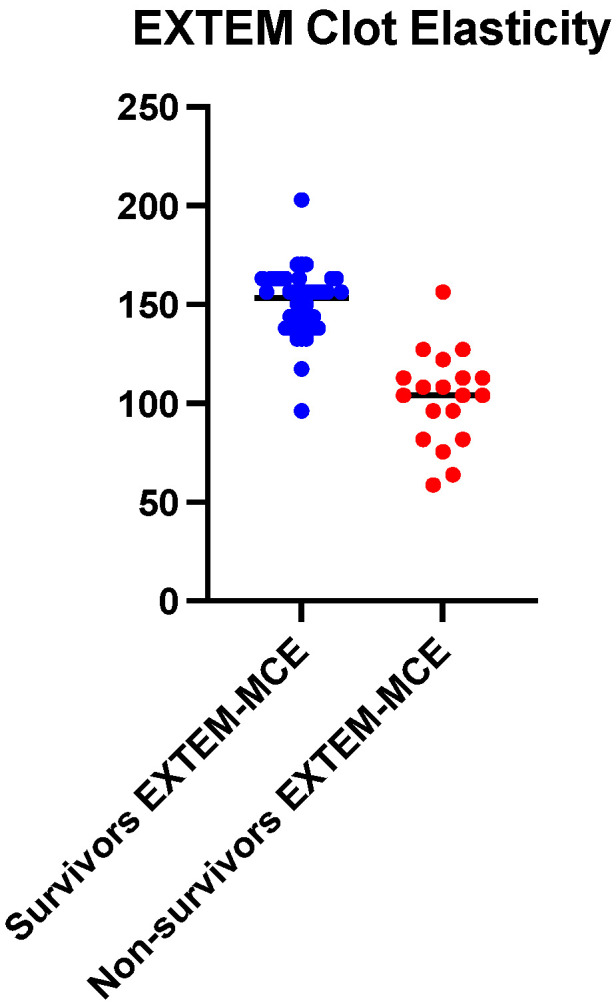
Mean values comparison of maximum clot elasticity (MCE) between survivors (n = 44) and non-survivors (n = 19), EXTEM channel.

**Figure 4 ijms-25-07763-f004:**
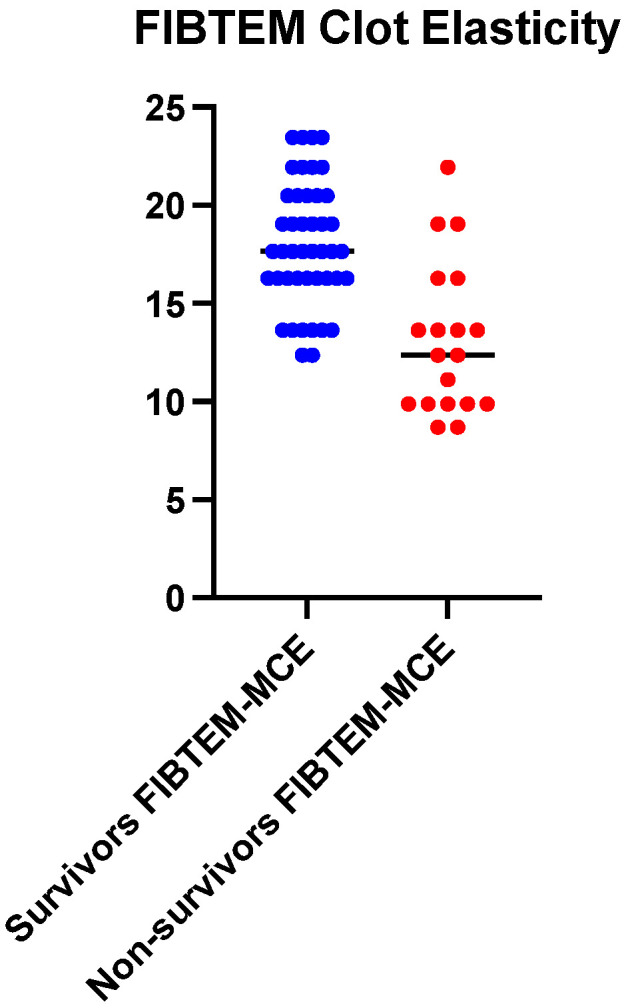
Mean values comparison of maximum clot elasticity (MCE) between survivors (n = 44) and non-survivors (n = 19), FIBTEM channel.

**Figure 5 ijms-25-07763-f005:**
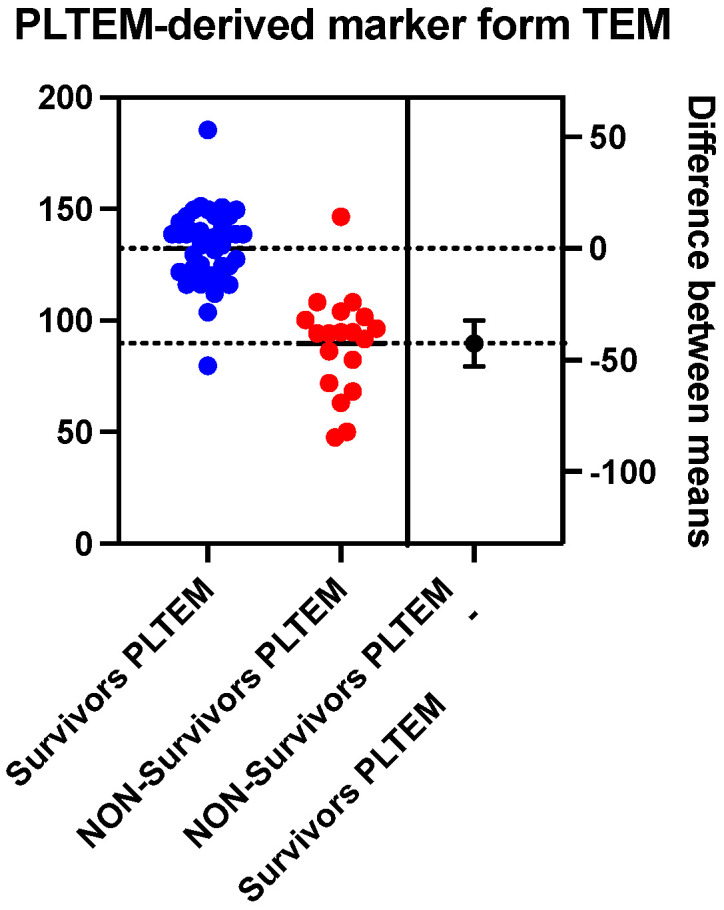
Mean values comparison (dashed lines) of platelet contribution (PLTEM derived marker) between survivors (n = 44) and non-survivors (n = 19).

**Figure 6 ijms-25-07763-f006:**
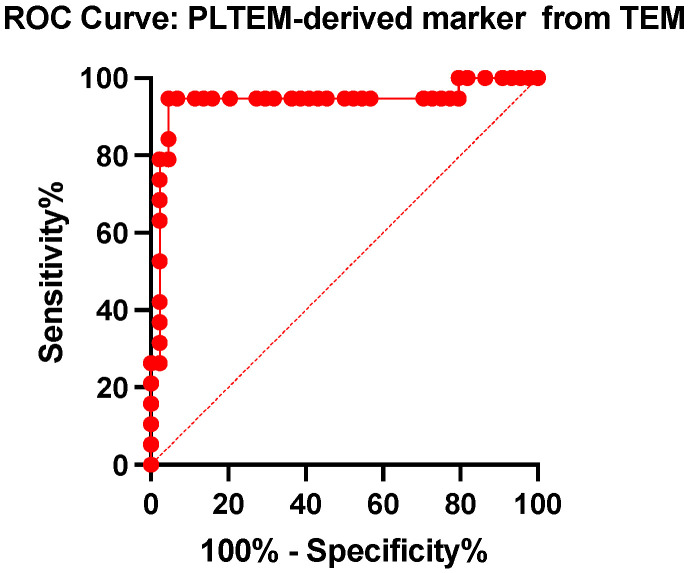
Receiver operating characteristics curve (ROC) of the PLTEM for predicting mortality in TBI patients.

**Figure 7 ijms-25-07763-f007:**
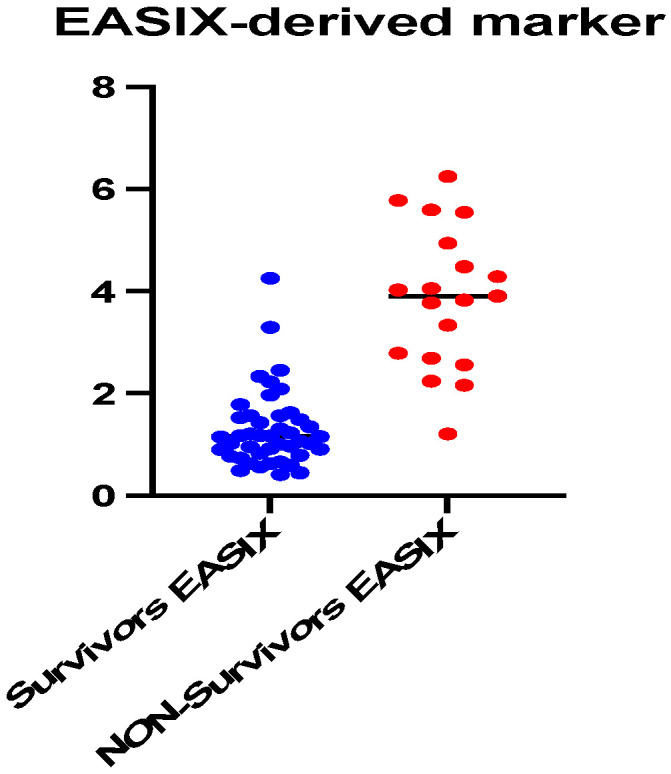
Median values comparison of endothelial activation and stress index (EASIX) between survivors (n = 44) and non-survivors (n = 19).

**Figure 8 ijms-25-07763-f008:**
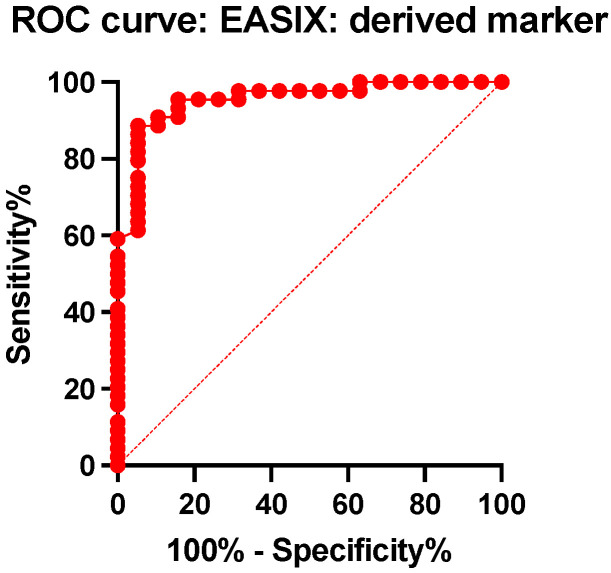
Receiver operating characteristics curve (ROC) of the EASIX for predicting mortality in TBI patients.

**Figure 9 ijms-25-07763-f009:**
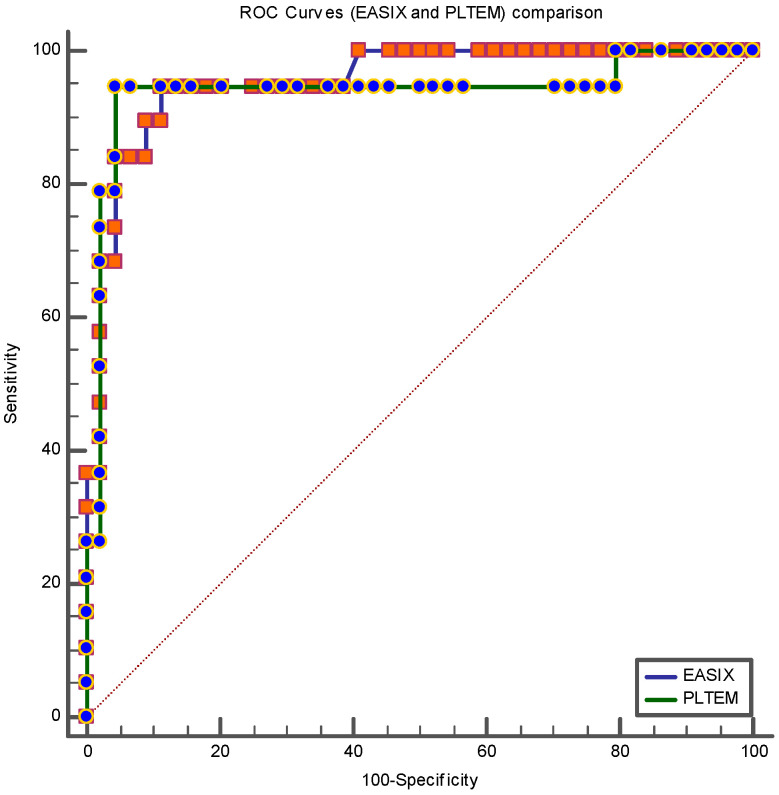
ROC curves comparison—PLTEM vs. EASIX—Classification variable: death at 30 days. Hanley and McNeil analysis.

**Figure 10 ijms-25-07763-f010:**
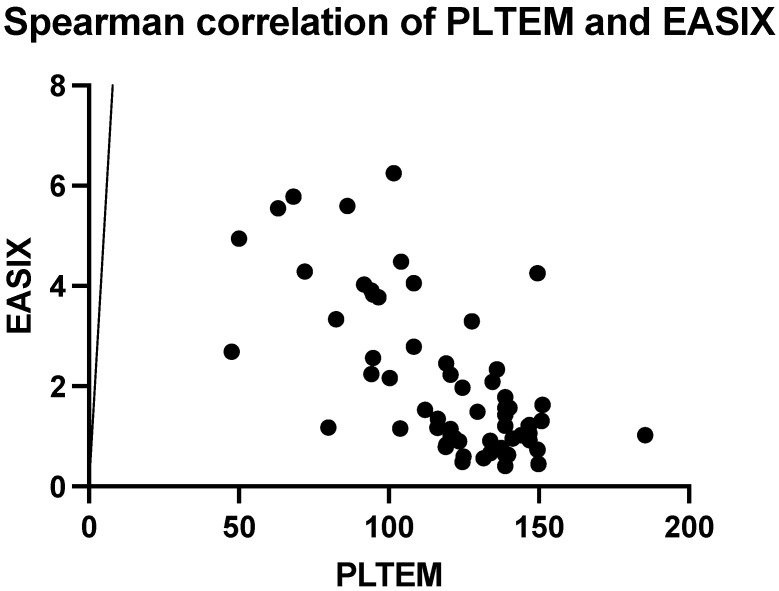
Spearman correlation of platelet contribution (PLTEM) and endothelial activation and stress index (EASIX).

**Table 1 ijms-25-07763-t001:** Comparative analysis between paraclinical data of survivors and non-survivors.

Parameters	Survivor (n = 44)	Non-Survivor (n = 19)	*p* Value—Two Tailed
Age (years) *	40.64 ± 15.04	44.95 ± 15.84	0.30
BMI (kg/m^2^) *	26.11 ± 4.20	25.12 ± 3.90	0.38
Severe TBI ***	n = 34 (77.27%)	n = 16 (84.21%)	0.73
Moderate TBI ***	n = 10 (22.73%)	n = 3 (15.79%)	
Epidural hematoma ***	7 (15.91%)	2 (10.53%)	0.71
Subdural hematoma ***	10 (22.73%)	8(42.11%)	0.13
Subarachnoid hemorrhage ***	26 (52.63%)	10 (59.09%)	0.78
pH *	7.28 ± 0.10	7.22 ± 0.15	0.07
Base excess (mmol/L) *	−7.03 ± 6.99	−11.90 ± 10.85	0.03
Serum lactate ** (mmol/L)	2.10 (1.1–3.4)	2.60 (0.6–6)	0.29
Serum bicarbonate (mmol/L) *	19.70 ± 4.11	20.00 ± 5.73	0.81
Hemoglobin (g/dL) **	9.60 (8.7–11.60)	9.30 (8.8–10.70)	0.72
Platelets/µL **	189,000 (143,000–298,000)	108,000 (89,000–151,000)	<0.0001
INR **	1.76 (1.11–2.46)	2.40 (1.22–3.56)	0.02
aPTT (sec) *	36.16 ± 7.40	39.11 ± 7.80	0.15
EXTEM A5 (mm) *	38.2 ± 3.18	28.95 ± 5.89	<0.0001
EXTEM A10 (mm) *	47.59 ± 3.28	38.11 ± 6.02	<0.0001
EXTEM A20 (mm) *	56.14 ± 3.40	45.37 ± 6.68	<0.0001
EXTEM MCF (mm) *	59.86 ± 2.80	50.05 ± 5.99	<0.0001
FIBTEM A5 (mm) *	9.61 ± 1.79	6.78 ± 2.41	<0.0001
FIBTEM A10 (mm) *	11.11 ± 1.94	7.94 ± 2.65	<0.0001
FIBTEM A20 (mm) *	13.52 ± 2.17	10.16 ± 2.63	<0.0001
FIBTEM MCF (mm) *	15.20 ± 2.25	11.53 ± 2.93	<0.0001
LDH (U/L) **	348.50 (303.00–423.5)	484.00 (411.00–624.00)	<0.0001
Serum creatinine (mg/dL) **	0.64 (0.60–0.73)	0.82 (0.74–0.82)	<0.0001

* Unpaired *t*-test for parametrical data (the means and standard deviations are presented in the table); ** Mann–Whitney test for non-parametrical data (the medians and interquartile ranges are presented in the table); BMI—body mass index; INR—international normalized ratio; aPTT—activated partial thromboplastin time; EXTEM A5, A10, A20—clot amplitude at 5, 10, and 20 min in EXTEM channel; EXTEM MCF—maximum clot firmness in EXTEM channel; FIBTEM A5, A1, A20—clot amplitude at 5, 10, and 20 min in FIBTEM channel; FIBTEM MCF—maximum clot firmness in FIBTEM channel; LDH—lactate dehydrogenase. *** Fisher’s exact test.

## Data Availability

All presented data are available on demand.

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
