# Peer review of "Platelet Contribution and Endothelial Activation and Stress Index-Potential Mortality Predictors in Traumatic Brain Injury"

_ijms, 2024, doi:10.3390/ijms25147763_

Round 1

Reviewer 1 Report

Comments and Suggestions for Authors

Title - I suggest a minor edit for readability - Platelet contribution (PLTEM) and "endothelial activation stress index score" (EASIX) as potential mortality predictors in traumatic brain injury

Abstract - The methods do not clearly indicate the "two studied groups" which are then mentioned in the results. Please clarify. (ROTEM) should be added after rotational thrombelastometric on Line 21 so the ROTEM abbreviation is understandable when it is mentioned in the results. Since the EASIX score is not well known a brief description should be provided in the methods on the clinical variables that are used to calculate it. Please clarify what parameter was used to determine platelet component (e.g., was it MCF or A10 or MCE as all have been used in the literature). Line 28 - please clarify what parameter the 114.10 mm is referring it (noting that platelet MCE does not have units which cancel out in the calculation).

Introduction: Line 44 - Please clarify if 60.611 should be 60,611 (i.e. 60 thousand six hundred and eleven). Line 45 - please clarify whether you are indicating that mortality is declining and provide literature evidence for this because prevalence/incidence of TBI is certainly not declining. Line 49 - it should be specified that this is prediction for TBI (or TBI of a certain severity if that is the case, e.g., severe). Line 74 - rather than BBB rupture, this would be more accurately described as BBB breach. Line 78 - rather than "ongoing shock" which may not apply to all TBI patients, particularly those with isolated injury, it would be more appropriate to use "injury" - activation of the SNS occurs immediately after injury, whether TBI or other. Line 113 - correct spelling of assesses. Please include a specific aim at the conclusion of the Introduction which accurately reflects the study presented.

Methods - Line 147- please capitalise European. Line 157 - in line with more scientific writing, I suggest rather than "We analysed the utmost relevant data", "The following data was analysed". Please justify the exclusion of analysis of ROTEM clot lysis parameters (and APTEM results if available) given the Introduction and previous literature on what is known about TBI-IC. Line 162 - please provide a reference for the calculation of EASIX. Line 163 - 9 should be superscripted in the units for platelet count. What was the justification for using INR and aPTT rather than EXTEM CT and INTEM/HEPTEM CT if available given the known limitations of these plasma-based assays. Specify whether quality controls were performed on the ROTEM Sigma as per manufacturer's instructions. Line 199 - please provide Emergency Clinical Hospital ethics committee approval number or reference. Further information should be included on the hospital level, e.g., is this a tertiary level hospital with ICU that provides neurocritical care? Is it a referral hospital and so are some patients transferred from other smaller clinics and hospitals which potentially delays the time to initial clinical and coagulation testing? Some context should be provided in terms of the average time to hospital for these patients given the time critical nature of TBI and the known effect of delayed hospital admission on mortality of msTBI patients. This is a potential limitation that should also be addressed in the Discussion.

Results - Severity of injury is very important in terms of mortality from TBI. The number of moderate TBI patients vs severe TBI patients in both survivor and non-survivor groups should be reported. Table 1 - please spell out all abbreviations in the Table legend. All figures should have more descriptive legends which include the n value per group and how data are presented. Line 241 - the mm units need clarification (as for the Abstract). Line 247-248 - provide measure of variance (e.g., IQR) for the EASIX results. Was analysis considered to compare the predictive nature of the PLTEM vs EASIX as a function of early (e.g. within the first 24 h after injury) vs later deaths given dynamics of TBI pathophysiology (with a potential hypothesis that PLTEM might be more associated with the early deaths? Is there a significant strong correlation between PLTEM and EASIX values? 

Discussion - Line 262 - The statement that progress has been made in reducing the global burden of TBI in recent years is not consistent with recent literature showing up to 70 million new cases annually and incidence increasing globally. Line 273 - resuscitation-induced coagulopathy is replicated with a mention in previous statement. Line 337 - Please correct spelling of Swedish and capitalise. Line 338 - please define MEA. Line 339 - please define AA. Further discussion/critical analysis should be included on why the authors believe PLTEM was a better mortality predictor than EASIX in this patient population. The discussion should also include the potential clinical translation of these markers, e.g., can PLTEM be easily calculated with other VHA devices, e.g., TEG, ClotPro and what about paediatric patients - would there be any differences that would potentially limit the use of these markers in paediatric TBI patients? How would these markers inform early treatment of TBI patients in a way that could reduce morbidity and mortality?

Conclusion - Lne 362 - the patient population should be restated, e.g., In conclusion, the present retrospective study indicates the potential of the parameters PLTEM and EASIX as markers of mortality prognosis in moderate-severe TBI patients, reflecting TBI-induced coagulopathy. 

Comments on the Quality of English Language

Minor English Editing required as indicated in the above comments.

Author Response

Dear reviewer,

We are pleased to resubmit for publication the revised version of manuscript no. ijms-3074014, entitled “Platelet contribution (PLTEM) and "endothelial activation stress index score" (EASIX) as potential mortality predictors in traumatic brain injury”. We appreciate the time and efforts you made in proofreading this manuscript. Your feedback helped improve our paper and clarify essential aspects. 

With reference to the comments and suggestions:

Comments and Suggestions for Authors

Title - I suggest a minor edit for readability - Platelet contribution (PLTEM) and "endothelial activation stress index score" (EASIX) as potential mortality predictors in traumatic brain injury

Thank you for the suggestion. Indeed, the title represents the one of the most important messages of the research for the majority of readers. We took your advice and simplified the title to “Platelet contribution and endothelial activation and stress index - potential mortality predictors in traumatic brain injury”.

Abstract

  • The methods do not clearly indicate the "two studied groups" which are then mentioned in the results. Please clarify→ The two groups were defined: survivors and non-survivors at 30 days after TBI.
  • (ROTEM) should be added after rotational thrombelastometric on Line 20 so the ROTEM abbreviation is understandable when it is mentioned in the results → Abbreviation was added.
  • Since the EASIX score is not well known a brief description should be provided in the methods on the clinical variables that are used to calculate it → EASIX formula was described.
  • Please clarify what parameter was used to determine platelet component (e.g., was it MCF or A10 or MCE as all have been used in the literature) → The contribution of platelets to clot strength is derived from maximum clot elasticity (MCE) and maximum clot firmness (MCF). MCE is defined as (MCFx100)/(100-MCF) and PLTEM as EXTEM MCE – FIBTEM MCE.
  • Line 28 - please clarify what parameter the 114.10 mm is referring it (noting that platelet MCE does not have units which cancel out in the calculation) → The ROTEM-derived CE is a “relative” measure of clot elasticity, expressed without units (i.e., dimensionless). The term maximum clot elasticity (MCE) at maximum amplitude is used with ROTEM. We have corrected the misused unit of measurement.

Introduction:

  • Line 49 - Please clarify if 60.611 should be 60,611 (i.e. 60 thousand six hundred and eleven) → We clarified as follows: ”more than 60 thousand TBI-related deaths were recorded in the USA”
  • Line 50 - Please clarify whether you are indicating that mortality is declining and provide literature evidence for this because prevalence/incidence of TBI is certainly not declining.→ We clarified that the overall mortality and global burden of disease in TBI is declining over a large period of time, that is in the last 30 years, mostly due to better understanding of the physiopathology and subsequent treatment, international guidelines implementantion, and health policies directly addressing this issue by traffic and work safety regulations. As stated in Guan’s “Global, regional and national burden of traumatic brain injury and spinal cord injury, 1990–2019: a systematic analysis for the Global Burden of Disease Study 2019”, used as reference [2] in our paper, ”Global age-standardised incidence rates of TBI decreased significantly by −5.5% (95% UI −8.9% to −3.0%) from 1990 to 2019”.
  • Line 54 - it should be specified that this is prediction for TBI (or TBI of a certain severity if that is the case, e.g., severe)→ We specified as follows: ”Mortality prediction for TBI was studied extensively.”
  • Line 79 - BBB rupture → BBB breach
  • Line 83- "ongoing shock" →"injury"
  • Line 120- correct spelling of “asseses”→ “assesses”
  • Please include a specific aim at the conclusion of the Introduction which accurately reflects the study presented.→ At your very welcomed suggestion, we highlighted that the ROTEM parameter PLTEM, alone or in association with the EASIX marker of endotheliopathy mirrors essential aspects of TBI-IC physiopathology, and could be used to create a TBI-IC prognosis tool to help in decision-making and reduction of the burden of disease by adequate allocation of resources in a personalized and timely manner (line 140).

Methods

  • Line 164- please capitalise European.→ The correction was made.
  • Line 173 - in line with more scientific writing, I suggest rather than "We analysed the utmost relevant data", "The following data was analysed".→ The correction was made according to your suggestion.
  • Please justify the exclusion of analysis of ROTEM clot lysis parameters (and APTEM results if available) given the Introduction and previous literature on what is known about TBI-IC

and

What was the justification for using INR and aPTT rather than EXTEM CT and INTEM/HEPTEM CT if available given the known limitations of these plasma-based assays.

→ For this study, we focused on the platelet and endothelial contribution to clot dynamics and will consider other ROTEM-derived parameters when analyzing other specific key components of clot formation and lysis, as we are aware of the limitations of conventional coagulation assays, that we also highlighted in our paper.

  • Line 177-179 - please provide a reference for the calculation of EASIX.→The reference was added
  • Line 179 - 9 should be superscripted in the units for platelet count.→ The correction was made
  • Specify whether quality controls were performed on the ROTEM Sigma as per manufacturer's instructions. → Periodic quality control for monitoring accuracy and precision of tests carried out on the ROTEM Sigma were performed as per manufacturer's instructions. (line 189-190)
  • Line 227-229 - please provide Bucharest Emergency Clinical Hospital ethics committee approval number or reference.→ Clinical Emergency Hospital of Bucharest Ethics Committee (protocol code 11095/03.12.2019 was added.)
  • Further information should be included on the hospital level, e.g., is this a tertiary level hospital with ICU that provides neurocritical care? Is it a referral hospital and so are some patients transferred from other smaller clinics and hospitals which potentially delays the time to initial clinical and coagulation testing? Some context should be provided in terms of the average time to hospital for these patients given the time critical nature of TBI and the known effect of delayed hospital admission on mortality of TBI patients. This is a potential limitation that should also be addressed in the Discussion.

            Thank you for the suggestion, we detailed these issues in both the materials and methods section of the exclusion criteria and the discussion section.

Results

  • Severity of injury is very important in terms of mortality from TBI. The number of moderate TBI patients vs severe TBI patients in both survivor and non-survivor groups should be reported.

            Thank you for the suggestion, we have updated Table 1 with the required information and a necessary comment in the results section

  • Table 1 - please spell out all abbreviations in the Table legend.→Abbreviations were added.
  • All figures should have more descriptive legends which include the n value per group and how data are presented. -we have added explanations under each graph
  • The mm units need clarification (as for the Abstract).→We corrected the misused unit of measurement.
  • Line 291-293 - provide measure of variance (e.g., IQR) for the EASIX results. We have added IQR for EASIX
  • Was analysis considered to compare the predictive nature of the PLTEM vs EASIX as a function of early (e.g. within the first 24 h after injury) vs later deaths given dynamics of TBI pathophysiology (with a potential hypothesis that PLTEM might be more associated with the early deaths?

            This idea is an extraordinary one, but unfortunately not applicable in our study, given the small number of patients included. In the group of deceased patients we have only one death in the first 24 hours, which makes a relevant statistical analysis impossible. However, the "validation” of PLTEM and EASIX as a prognostic marker of general mortality in moderate and severe traumatic brain injury in several studies, possibly multicenter, will certainly lead to the deepening of their selective predictive power. Given the similarity of the ROC curves, but the superiority of PLTEM in the Cox-Proportional Hazard ratio, we raised a possible hypothesis in the discussion section related to the use of PLTEM for the detection of early deaths. Line 393-409

  • Is there a significant strong correlation between PLTEM and EASIX values? 

Thank you for the suggestion, we found a strong correlation between PLTEM and EASIX. We have added all the details in the results section. Line 306-312

Discussion

  • Line 317 - The statement that progress has been made in reducing the global burden of TBI in recent years is not consistent with recent literature showing up to 70 million new cases annually and incidence increasing globally.→ At your suggestion, we addressed this in the Introduction section.
  • Resuscitation-induced coagulopathy is replicated with a mention in previous statement.→ We added a clarifying introduction paragraph: Resuscitation-associated endotheliopathy (RAsE) and subsequent coagulopathy, mostly through aggressive volume-expansion resuscitation, may further aggravate SHINE. Line 86-88
  • Line 413 - Please correct spelling of Swedish and capitalise → We made the correction
  • Line 338 - please define MEA→ We defined MEA on line 109 - multiple electrode aggregometry
  • Line 339 - please define AA → We defined AA on line 98 - arachidonic acid
  • Further discussion/critical analysis should be included on why the authors believe PLTEM was a better mortality predictor than EASIX in this patient population.

            In our study, Cox-Proportional Hazard ratio regression showed PLTEM to have the greatest influence on survival time after moderate/severe TBI, followed by EASIX, even though ROC curves were pretty similar. ROTEM-derived platelet contribution (PLTEM) to clot formation showed a great predictive value for mortality prediction (death in the first 30 days after injury) with a cut-off point of 114.11, 94.74% sensitivity, and 93.18% specificity. The predictive performance of EASIX was quite similar with PLTEM, with a cut-off-point of 2.12, a sensitivity of 88.64%, and a specificity of 94.74%. EASIX integrates indirect markers of endothelial dysfunction, including platelet count, since low platelet levels may be attributable to hyperactivation and hyperaggregation.

            EASIX embodies the endothelial component of coagulopathy itself, but also the subsequent endotheliopathy-associated thrombocytopenia. From the best of our knowledge this is the first study to confirm this pathophysiological link through a direct correlation analysis (r= -0,57, p<0.0001) for moderate/severe traumatic brain injury patients. It still remains to be established why PLTEM seems to better predict the hazard of death, if the ROC curve for mortality prediction is similar to that of EASIX. A possible answer, which we cannot demonstrate in this unicentric study with a small number of patients, could be the possible involvement of PLTEM in the prediction of early-deaths of moderate/severe traumatic brain injury patients.

            -added to discussion line 393-409

  • The discussion should also include the potential clinical translation of these markers, e.g., can PLTEM be easily calculated with other VHA devices, e.g., TEG, ClotPro and what about paediatric patients - would there be any differences that would potentially limit the use of these markers in paediatric TBI patients?

→Multiple literature reviews investigating TEG and ROTEM in trauma concluded that the VHA’s results are not interchangeable, although they are similar in clinical applicability, diagnosis, treatment, and prognosis. The most cited possible explanation for this phenomenon is the different technique itself, with different clotting activators and physical and chemical properties (1,2)

Emergency Clinical Hospital Bucharest only provides health care to adult patients, therefore we cannot perform a comparative study between adult and pediatric TBI patients. However, the literature search showed that TEM-guided blood component replacement is an emerging practice in children for both traumatic and non-traumatic bleeding, with high levels of fibrinolysis across different injuries, perhaps reflecting distinct pathophysiological pathways or injury patterns in children compared to adults, that are worth further investigation (3,4). We are not aware of PLTEM or EASIX analysis in pediatric TBI patients.       

  1. Sandro Rizoli, Arimie Min, Adic Perez Sanchez, Pang Shek, Richard Grodecki, PrecillaVeigas, Henry T. Peng, In Trauma, Conventional ROTEM and TEG Results Are            Not Interchangeable But Are Similar in Clinical Applicability, Military Medicine,             Volume 181, Issue suppl_5, May 2016, Pages 117–126,       https://doi.org/10.7205/MILMED-D-15-00166
  2. Sankarankutty, A., Nascimento, B., Teodoro da Luz, L. et al. TEG® and ROTEM® in             trauma: similar test but different results?. World J Emerg Surg 7 (Suppl 1), S3 (2012). https://doi.org/10.1186/1749-7922-7-S1-S3
  3. George, S., Wake, E., Sweeny, A., Campbell, D. and Winearls, J. (2022), Rotational thromboelastometry in children presenting to an Australian major trauma centre: A     retrospective cohort study. Emergency Medicine Australasia, 34: 590-598.             https://doi.org/10.1111/1742-6723.13939
  4. Deng Q, Hao F, Wang Y, Guo C. Rotation thromboelastometry (ROTEM) enables improved outcomes in the pediatric trauma population. J Int Med Res. 2018;46(12):5195-          5204. doi:10.1177/0300060518794092

  • How would these markers inform early treatment of TBI patients in a way that could reduce morbidity and mortality?

            This question could be the essence of a much larger future research project. We really appreciate the fact that you also thought about this aspect. Such  thoughts will keep medical research alive.

            The treatment of TBI-induced coagulopathy remains on the one hand with some strict recommendations and on the other hand it will have continuous changes depending on the advances in understanding the physiopathology. PLTEM and EASIX comprise a coagulation part as far as we know to date. Before we can guide a treatment, or make any kind of recommendation, these derived markers will need extensive validation. At the moment, through this study, we only want to bring the two parameters together. We are not prepared to make any assumptions or recommendations regarding any treatment. (line 444-449)

  • Line 362 - the patient population should be restated, e.g., In conclusion, the present retrospective study indicates the potential of the parameters PLTEM and EASIX as markers of mortality prognosis in moderate-severe TBI patients, reflecting TBI-induced coagulopathy.→ At your very welcomed  suggestion, we highlighted the potential of the parameters PLTEM and EASIX as markers of mortality prognosis, reflecting TBI-IC in patients with moderate and severe TBI.

Reviewer 2 Report

Comments and Suggestions for Authors

This research is retrospective analysis of small number of traumatic patients.

Only survival was used as the outcome.

The patients with severe systemic damage were excluded. There is no description about minor systemic damage and nature of brain injury.

Author used many abbreviations without definitions even in title and abstract. All abbreviation should be fully defined at first appearance and should be used after the definition. You should not use abbreviation if you use the term once.

Author Response

Dear reviewer,

We are pleased to resubmit for publication the revised version of manuscript no. ijms-3074014, entitled “Platelet contribution (PLTEM) and "endothelial activation stress index score" (EASIX) as potential mortality predictors in traumatic brain injury”. We appreciate the time and efforts you made in proofreading this manuscript. Your feedback helped improve our paper and clarify essential aspects. 

With reference to the comments and suggestions:

Comments and Suggestions for Authors

  • This research is retrospective analysis of small number of traumatic patients. Only survival was used as the outcome.

Indeed, this is a major limitation of our study. The promising results made us seriously consider designing a study methodology for a prospective multicentric study on TBI-IC including all Emergency Hospitals in Bucharest and using multiple end-points. We would like to investigate whether PLTEM and EASIX could be integrated into multiple-variable prognostication tools and whether they show the same relevance in more complex neurological outcome prediction for TBI (evolution to brain death, cerebral performance and cognitive status, functional outcome, quality of life, etc.), not only 30-day survival. (updated in discussion section line 424-431)

  • The patients with severe systemic damage were excluded. There is no description about minor systemic damage and nature of brain injury.

            Thank you for this observation. We have tried to highlight the distinct coagulopathy associated with TBI, especially the endothelial response. Therefore, including multiple trauma patients who also had severe or moderate TBI would make it difficult to accurately interpret the data on endothelium and platelet contribution to clot dynamics, especially since we couldn’t use a molecular marker of endothelial damage, such as syndecan-1, nor could we distinguish between local, cerebral and systemic response.

            In our group of patients, the Abbreviated Injury Scale (AIS) score was a maximum of 5 (critical lesion) for the Head and neck worst injury and a maximum of 2 (moderate lesion) in any other two body systems. -added on materials and methods section. (updated in materials and methods section line 147-149).

  • Author used many abbreviations without definitions even in title and abstract. All abbreviation should be fully defined at first appearance and should be used after the definition. You should not use abbreviation if you use the term once.

Thank you for the patience and attention you put into underlining these aspects. They have all been addressed. We took your advice and simplified the title to “Platelet contribution and endothelial activation and stress index - potential mortality predictors in traumatic brain injury”.

Reviewer 3 Report

Comments and Suggestions for Authors

Dear Authors,

I have reviewed your manuscript titled "Platelets contribution (PLTEM) and 'endothelial activation stress index score' (EASIX) - potential mortality predictors in traumatic brain injury." I found the topic interesting, and I appreciate the effort put into this research. However, I have several suggestions to improve the clarity and readability of your manuscript:

  1. Introduction: While the introduction provides a good overview, it lacks a clear emphasis on the main findings of your study. It is essential to make these findings more explicit to highlight the importance of your work.

  2. Materials and Methods:

    • Clearly list all the statistical methods used, including the Two-Way ANOVA, which is mentioned in the results but not described in the methods section.
    • Provide more details on the criteria for performing ROTEM analysis in the patients. Specify who decided to perform ROTEM, the time points for these analyses, and whether other paraclinical parameters were measured simultaneously.
  3. Results:

    • Explain graphs 1 and 2 more clearly in the results section and provide detailed explanations under each graph.
    • Clarify the parameters used in the Cox regression analysis to ensure that readers understand the basis of your conclusions.
  4. Discussion:

    • Emphasize the main findings of your study in the first paragraph of the discussion. Highlight the significance of these findings and their potential impact on the field.

Author Response

Dear reviewer,

We are pleased to resubmit for publication the revised version of manuscript no. ijms-3074014, entitled “Platelet contribution (PLTEM) and "endothelial activation stress index score" (EASIX) as potential mortality predictors in traumatic brain injury”. We appreciate the time and efforts you made in proofreading this manuscript. Your feedback helped improve our paper and clarify essential aspects. 

With reference to the comments and suggestions:

Reviewer 3

Dear Authors,

I have reviewed your manuscript titled "Platelets contribution (PLTEM) and 'endothelial activation stress index score' (EASIX) - potential mortality predictors in traumatic brain injury." I found the topic interesting, and I appreciate the effort put into this research. However, I have several suggestions to improve the clarity and readability of your manuscript:

Introduction

While the introduction provides a good overview, it lacks a clear emphasis on the main findings of your study. It is essential to make these findings more explicit to highlight the importance of your work.

At your very welcomed suggestion, we highlighted our working hypothesis that the ROTEM parameter PLTEM could be used as a dynamic tool and a marker of TBI-IC and mortality prognosis instrument in TBI patients. (line 139-142)

Materials and Methods

Clearly list all the statistical methods used, including the Two-Way ANOVA, which is mentioned in the results but not described in the methods section.

We mentioned in the Materials and Methods section the Two-Way-Anova test that we used to determine how the clot amplitude changes quantitatively in time for the two groups studied. (line 223-225)

Provide more details on the criteria for performing ROTEM analysis in the patients. Specify who decided to perform ROTEM, the time points for these analyses, and whether other paraclinical parameters were measured simultaneously.

All patients had complete blood work analysis (including complete blood count, classical coagulation assay, and ROTEM analysis) performed in the Emergency Department. Initially, if needed, patients received liberal blood transfusions, but after obtaining the results of the first thromboelastometric assay, they were switched to a goal-directed algorithm. When all the results were available, the patient was usually in the ICU. In individual cases (severe coagulopathy, more than two hours delay between ED presentation and ICU admission, or at clinician’s discretion), classical coagulation and ROTEM assays were repeated at ICU admission. However, we didn’t elaborate on this since we only used first, singular admission data for all the patients included in the study.(line 186-187)

Results:

Explain graphs 1 and 2 more clearly in the results section and provide detailed explanations under each graph.

The variance detected for the EXTEM channel of the clot amplitude in the two groups is 26.59%, p<0.0001, while the variance for the FIBTEM detected between the groups of deceased and surviving patients of the clot amplitude at 5, 10, and 20 minutes, but also the MCF is 39.66%, p=0,003, as presented in Graphics 1 and 2. We have added explanations under each graphic.

Clarify the parameters used in the Cox regression analysis to ensure that readers understand the basis of your conclusions.

Thank you for helping us to highlight this important aspect. We discussed into detail in the dedicated section the Cox-Proportional Hazard ratio regression that showed PLTEM to have the greatest influence on survival time after moderate/severe TBI, followed by EASIX, and with ROC curves pretty similar. ROTEM-derived platelet contribution to clot formation PLTEM showed a great predictive value for mortality prediction (death in the first 30 days after injury) with a cut-off point of 114.11, 94.74% sensitivity, and 93.18% specificity. The predictive performance of EASIX was quite similar with PLTEM, with a cut-off-point of 2.12, a sensitivity of 88.64%, and a specificity of 94.74%. It still remains to be established why PLTEM seems to better predict the hazard of death, if the ROC curve for mortality prediction is similar to that of EASIX. A possible answer, which we cannot demonstrate in this unicentric study with a small number of patients, could be the possible involvement of PLTEM in the prediction of early-deaths of moderate/severe TBI patients. (line 314-322) and (line 393-409)

Discussion:

Emphasize the main findings of your study in the first paragraph of the discussion. Highlight the significance of these findings and their potential impact on the field.

Prognostication is essential in giving families and healthcare personnel reasonable expectations that aid in decision-making in severe, debilitating diseases, such as TBI. Simultaneously, prognostication helps reduce the global burden of disease by adequate allocation of resources in a personalized manner. We demonstrated that specific thromboelastometry-derived parameters that reflect platelet contribution to clot formation - PLTEM, alone or in association with markers of endotheliopathy - EASIX mirror essential aspects of TBI-IC physiopathology, and could be used to create a TBI-IC prognosis tool. Further prospective studies conducted in larger centers are needed to investigate a combined score for mortality prediction in TBI using PLTEM and EASIX. (line 330-338).

Round 2

Reviewer 1 Report

Comments and Suggestions for Authors

Thank you to the authors for addressing all previous comments and improving the manuscript.

Reviewer 2 Report

Comments and Suggestions for Authors

I recognized significant improvements of this manuscript.

Reviewer 3 Report

Comments and Suggestions for Authors

Dear Authors,

I have reviewed the revised version of your manuscript. I appreciate the thoroughness with which you addressed my previous suggestions. I am pleased to suggest accepting the revised version of your manuscript. Congratulations on your excellent work.

Best regards,